# Observation of non-superconducting phase changes in nitrogen doped lutetium hydrides

Xiangzhuo Xing [1,2,9], Chao Wang[1,2,9], Linchao Yu[1], Jie Xu[1], Chutong Zhang[1], Mengge Zhang[1], Song Huang[1], Xiaoran Zhang[1], Yunxian Liu[1,2], Bingchao Yang[1,2], Xin Chen [1,2], Yongsheng Zhang[1,2], Jiangang Guo [3], Zhixiang Shi [4], Yanming Ma [5,6,7], Changfeng Chen[8] & Xiaobing Liu [1,2] ✉

The recent report of near-ambient superconductivity and associated color changes in pressurized nitrogen doped lutetium hydride has triggered worldwide interest and raised major questions about the nature and underlying physics of these latest claims. Here we report synthesis and characterization of high-purity nitrogen doped lutetium hydride $LuH_{2\pm x}N_y$. We find that pressure conditions have notable effects on Lu-N and Lu-NH chemical bonding and the color changes likely stem from pressure-induced electron redistribution of nitrogen/vacancies and interaction with the $LuH_2$ framework. No superconducting transition is found in all the phases at temperatures 1.8-300 K and pressures 0-38 GPa. Instead, we identify a notable temperature-induced resistance anomaly of electronic origin in $LuH_{2\pm x}N_y$, which is most pronounced in the pink phase and may have been erroneously interpreted as a sign of superconducting transition. This work establishes key benchmarks for nitrogen doped lutetium hydrides, allowing an in-depth understanding of its novel pressure-induced phase changes.

A recent study reported near-ambient superconductivity in a nitrogen doped lutetium hydride (Lu-N-H in short)[1], which has triggered worldwide interest and raised major questions[2–7]. High-temperature superconductivity has been predicted even realized in metallic hydrogen and hydrogen-rich compounds (such as sulfur hydride[8], rare-earth hydrides[9–11], and alkaline-earth hydrides[12,13]), but megabar pressures are required to stabilize structures. The superconductivity in nitrogen doped lutetium hydride was claimed at much reduced pressures of 0.3-3 GPa, with maximum critical temperature $T_c = 294$ K at 1.3 GPa[1]. This discovery has sparked tremendous interest in the scientific community and beyond, ensuing studies quickly followed[14–37], which have thus far found no evidence supporting superconductivity in Lu-N-H systems.

It was reported that the presence of near-ambient superconductivity in the nitrogen doped lutetium hydride coincided with a visual color change of the sample from blue (0-0.3 GPa) through pink (0.3-3 GPa) to red (>3 GPa)[1]. Conspicuously, the superconducting state was only found in the pink phase. This is an intriguing phenomenon since all previously reported high-temperature superconductors show dark or black color in hydrogen-rich metallic systems[9–13]. Meanwhile, similar color changes were observed in $LuH_2$ at pressures ranging from 2.5 GPa to 5 GPa[14,26], but absent in $LuH_{2\pm x}N_y$ samples at pressures of 0-6.5 GPa[15,29]. No superconducting transition was detected in $LuH_2$ sample[14] or $LuH_{2\pm x}N_y$ sample up to 40.1 GPa[15,28] at temperatures from 300 K to 10 K. Since all the samples in the

[1]Laboratory of High Pressure Physics and Material Science (HPPMS), School of Physics and Physical Engineering, Qufu Normal University, Qufu 273165, China. [2]Advanced Research Institute of Multidisciplinary Sciences, Qufu Normal University, Qufu 273165, China. [3]Beijing National Laboratory for Condensed Matter Physics, Institute of Physics, Chinese Academy of Sciences, Beijing 100190, China. [4]School of Physics, Southeast University, Nanjing 211189, China. [5]Innovation Center for Computational Methods & Software, College of Physics, Jilin University, Changchun 130012, China. [6]State Key Laboratory of Superhard Materials, Jilin University, Changchun 130012, China. [7]International Center of Future Science, Jilin University, Changchun 130012, China. [8]Department of Physics and Astronomy, University of Nevada, Las Vegas, NV 89154, USA. [9]These authors contributed equally: Xiangzhuo Xing, Chao Wang. ✉e-mail: xiaobing.phy@qfnu.edu.cn

reported works share the same crystal structure, their main distinctions are the sample color and nitrogen contents. Ming *et al.* detected nitrogen element at four locations in ten measurements as 0.07, 0.12, 0.19, and 1.38 *wt.%* in the $LuH_{2\pm x}N_y$ samples[15], while the claimed near-ambient superconductor was reported in lutetium hydrides with 0.8–0.9 *wt.%* nitrogen doping[1]. Since the superconductivity in hydrides may be highly sensitive to nitrogen doping[38], producing and probing high-quality lutetium hydrides with well calibrated nitrogen doping is crucial to elucidating pertinent phenomena and the underlying mechanisms.

In this work, we synthesized pure bulk samples of nitrogen doped lutetium hydrides by high pressure and high temperature (HPHT) method. The samples exhibit uniform shinning blue color and have the same crystal structure and well-distributed nitrogen content as in previously reported samples[1]. Our in-situ high-pressure experiments revealed reversible color changes of nitrogen doped lutetium hydrides from shining blue to dark blue to purple and pink, finally into red. The critical pressures for these color changes are sensitive to the used pressure media, but the overall trends are consistent among all the cases. Our electrical measurements did not detect any signals of a superconducting transition at temperatures from 300 K to 1.8 K under pressures up to 38 GPa. There is an abnormal feature above 200 K during the warming-up electrical measurements on the purple, pink, and red phases, but the sample retains metallic behavior during the cooling-down measurements. The raw data taken in the warming-up measurements of the reported near-ambient superconductors[1] exhibit the same anomalous resistance, which was interpreted as a sign of superconductivity after a background subtraction was applied. Our results from both warming-up and cooling-down cycles on the same sample show unambiguous evidence that there is no correlation between the pink phase and the claimed near-ambient superconductivity in nitrogen doped lutetium hydrides since otherwise the data obtained during the cooling-down cycle also should capture the same basic physics associated with the superconducting transition. Also, the fact the same resistance anomaly is also seen in the red phase, which was recognized to be non-superconducting[1], further reinforces our conclusion about the lack of a superconducting transition in the sample.

## Results and discussion

### Synthesis, structure and composition of the produced $LuH_{2\pm x}N_y$ sample

We performed HPHT experiments for synthesis of nitrogen doped lutetium hydrides (see Methods for details and a schematic diagram in Supplementary Fig. 1). Figure 1a shows the X-ray diffraction (XRD) pattern on a polished smooth surface of the shining blue sample (top inset) by a diffractometer with wavelength of 1.5406 Å. All the dominant peaks can be well indexed as a cubic structure with $Fm\overline{3}m$ space group, and no impurity phases such as $LuN_{1-\delta}H_\varepsilon$ and $Lu_2O_3$ phases are detected except for a small amount of lutetium left during the surface polishing in preparation for XRD measurements. The crystal structure is consistent with that of $LuH_2$[39] and the reported near-ambient superconductors[1]. The sharp diffraction peaks indicate a better crystalline quality in our HPHT products compared to previous samples[1].

High-resolution transmission electron microscopy (HRTEM) measurements (Fig. 1b) reveal that the Lu atoms arrange in the (200), (111), and $(1\overline{1}\overline{1})$ orientations with the lattice spacing of 0.252 nm, 0.288 nm, and 0.289 nm, respectively. The selected area electron diffraction (SAED) pattern along the $[1\overline{1}0]$ zone-axis is shown in Fig. 1c, which is consistent with the result of our XRD measurement. The lattice parameter $a$ is determined to be 5.040 Å, which is much smaller than that of $LuH_3$[14], but is comparable to that of recently reported $LuH_{2\pm x}N_y$ samples[15]. Thus, we attribute the composition of our HPHT produced samples as $LuH_{2\pm x}N_y$.

Energy dispersive X-ray spectroscopy (EDX) spectrum in Fig. 1d shows clear evidence for the incorporation of nitrogen in our $LuH_{2\pm x}N_y$ samples, while the EDX mapping results indicate macroscopically uniform nitrogen distribution (Fig. 1e). We calculated the averaged nitrogen content in our $LuH_{2\pm x}N_y$ sample-I based on 10 randomly spots and four EDX mapping results, containing low nitrogen contents with an averaged value of 0.2 *wt.%*. For sample-II, we randomly measured 15 spots, obtaining an averaged nitrogen content of 0.84 *wt.%*, at the same level with previously reported nitrogen doped lutetium hydrides (0.8–0.9 *wt.%*)[1], indicating that these samples have similar nitrogen content. Details are given in Supplementary Tables 1 and 2.

Figure 1f displays typical Raman spectra of the $LuH_{2\pm x}N_y$ samples with a 532 nm laser excitation. The obtained spectrum of sample-II is practically identical to that of the previously reported sample[1] with characteristic peaks at 124 $cm^{-1}$, 147 $cm^{-1}$, 194 $cm^{-1}$, 251 $cm^{-1}$, and 1206 $cm^{-1}$, while all the Raman peaks in the sample-I has a nearly uniform down-shift of about 5 $cm^{-1}$. The peaks at 194 $cm^{-1}$, 251 $cm^{-1}$, and 1206 $cm^{-1}$ are close to those in $LuH_2$[14], thus likely coming from Lu-H framework or vacancy-associated vibration[25], while the peaks at the lower wavenumbers of 124 $cm^{-1}$ and 147 $cm^{-1}$ are only observed in nitrogen doped lutetium hydrides that can be assigned to Lu-NH and Lu-N related vibrational modes (Supplementary Figs. 2–4). No Raman signals of LuN phase[40] were detected in our produced samples. The scanning electron microscope and mapping measurements by Raman spectroscopy show a uniform polycrystalline structure of the produced $LuH_{2\pm x}N_y$ samples (Supplementary Figs. 5 and 6). X-ray photo-electron spectroscopy (XPS) measurements further confirm incorporation of nitrogen in the produced samples. Figure 1g shows that the main peak of the N 1$s$ core level is located at 399.64 eV. This suggests that the dominant bonding configuration for nitrogen atoms and NH species is N-H bonding in the produced $LuH_{2\pm x}N_y$ samples[41–43]. The shoulder seen at the lower binding energy of 398.68 eV indicates that some nitrogen atoms partially bond with lutetium, which is close to the value observed in hydrogen-doped lutetium nitrides (Supplementary Fig.7).

### Pressure-induced color changes in the $LuH_{2\pm x}N_y$ sample

To investigate the color evolution of the nitrogen doped lutetium hydrides, we loaded the produced $LuH_{2\pm x}N_y$ samples into standard diamond anvil cells without any medium or with different pressure media of nitrogen gas, water, silicon oil, and solid NaCl in the sample chambers. Our experimental results demonstrate that the color change with increasing pressure can be clearly seen in all the studied stress environments (Supplementary Figs. 8–13).

Figure 2a shows a systematic evolution pattern of color change of the $LuH_{2\pm x}N_y$ sample in the silicon oil medium over a wide pressure range. With pressure increasing to 6.2 GPa, the initial shining blue color turns to royal blue, then to purplish blue at 8.2 GPa, further to purple at 10.5 GPa. The intriguing pink phase starts to show up at 16 GPa and remains over a span of about 7 GPa until the vivid red color appears at 23.5 GPa. During the decompression process, the color evolution is completely revisable, gradually from vivid red back to its original shinning blue after pressure is fully released. Figure 2b depicts the critical pressure and stable region for the color changes in $LuH_{2\pm x}N_y$ samples in different pressure media. Our work reveals a previously unseen purple phase that is present in all the cases studied in our work, but was missed by Dasenbrock-Gammon et al. because of the very narrow pressure range for the transition from blue phase to pink phase in their work[1]. Our results establish a pressure-induced phase change sequence as indicated by their colors from phase I (blue) to phase II (purple), then phase III (pink), and eventually phase IV (red).

The overall trend of pressure induced color variation in our work is fully consistent with that of the reported near-ambient superconductors[1], but the critical pressures for color changes are higher in our work. This phenomenon is also observed in another

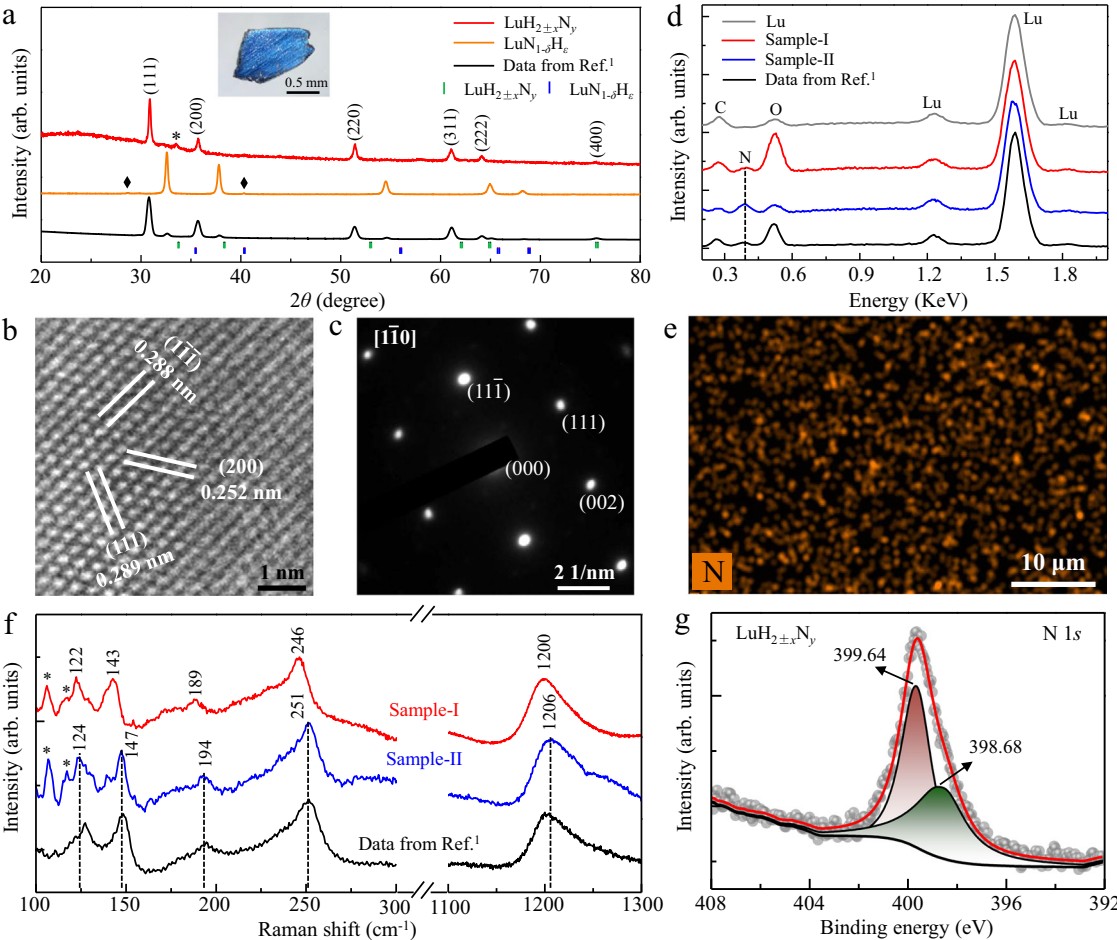

**Fig. 1 | Structure and composition of the produced nitrogen doped lutetium hydride sample.** The reported data of the near-ambient superconducting sample is plotted as black lines (bottom) for comparison. **a** Typical XRD pattern (red line) of our produced nitrogen doped lutetium hydride sample $LuH_{2\pm x}N_y$. The small peak marked by asterisk is from the residual lutetium. Top inset image is the measured bulk sample, around 1 mm in size. The XRD data of produced $LuN_{1-\delta}H_\varepsilon$ bulk sample (orange line) is also listed for comparison, in which the tiny peaks marked by rhombus come from unknown phase. **b, c** HRTEM image of the $LuH_{2\pm x}N_y$ sample and the SAED pattern along the $[1\bar{1}0]$ zone-axis. **d** Representative EDX spectra of the produced $LuH_{2\pm x}N_y$ sample-I (red line) and sample-II (blue line). The spectrum of the Lu piece used in the starting materials is also shown for comparison. The carbon peaks are from the tape used for holding a tiny sample for EDX measurements. **e** A typical EDX mapping image for nitrogen elements at four different areas, $50 \times 35$ μm in size. **f** Typical Raman spectra of $LuH_{2\pm x}N_y$ sample-I (red line) and sample-II (blue line) under ambient conditions. The peaks below 120 cm$^{-1}$ are from the background. **g** Typical XPS spectrum of the N 1$s$ core level.

independent work, where a dark-blue to pink-red color change occurs at pressure region of 11-21 GPa[15]. The much lower critical pressures (from 0.3 GPa to 3 GPa) in the previous work[1] can be attributed to the composite nature of their samples containing dominant N-doped lutetium hydride and minority $LuN_{1-\delta}H_\varepsilon$ phase with different lattice parameters, and these mixed phase structures can introduce large and complex internal stresses between the constituent components under compression. Such complex stresses could lead to significant changes, including large reduction of critical pressure for phase transitions[44–47]. This also explains the distinct critical pressures and stable regions of the differently colored phase in different pressure media that exert different stresses on the samples. To further verify this point, we introduce stresses by adding a small amount of nanodiamonds into $LuH_{2\pm x}N_y$ powers in the NaCl pressure medium, as illustrated in Fig. 2c. Consequently, the pink and red phases occur at significantly reduced pressures of 5.1 GPa and 10.0 GPa, respectively. This suggests that the presence of minority $LuN_{1-\delta}H_\varepsilon$ phase, which doesn't show any color change (Supplementary Fig. 14), mixed within dominant N-doped lutetium hydride in the samples of the previously report work[1] is effective in pushing the critical pressure for the pink phase down to 0.3 GPa.

## In-situ Raman spectrum of the $LuH_{2\pm x}N_y$ sample under compression

We investigated the pressure-dependent Raman spectrum of the produced $LuH_{2\pm x}N_y$ sample up to 30.75 GPa in the nitrogen gas medium, which has gone through the whole color change sequence (Fig. 3a and Supplementary Fig. 15). The strongest 1206 cm$^{-1}$ peak (Fig. 1f) overlap at relatively low pressures with the Raman peaks at 1332 cm$^{-1}$ from diamond anvils under compression, so we selected other characteristic peaks of 124 cm$^{-1}$, 147 cm$^{-1}$, 194 cm$^{-1}$, and 251 cm$^{-1}$ for comparison. The measured spectra are deconvoluted to analyze multipeak information[1]. Results in Fig. 3b show that all of these peaks gradually move upward in frequency at nearly the same rate, and the rate of decline decrease when it gets into the purple, pink and red phases. It is clearly seen in Fig. 3c that there is an accelerated rising rate for the intensity of N related peaks (124 cm$^{-1}$ and 147 cm$^{-1}$) when getting into the pink phase region and the intensity suddenly drop before the appearance of the red phase, while the Lu-H related peaks of 194 cm$^{-1}$ and 251 cm$^{-1}$ gradually decrease intensity at rising pressure through the purple and pink phases. The intensity of all peaks is more stable in the blue and red phases. It is interesting to note that the pressure driven changes of the intensity for Lu-N and Lu-NH related peaks in the

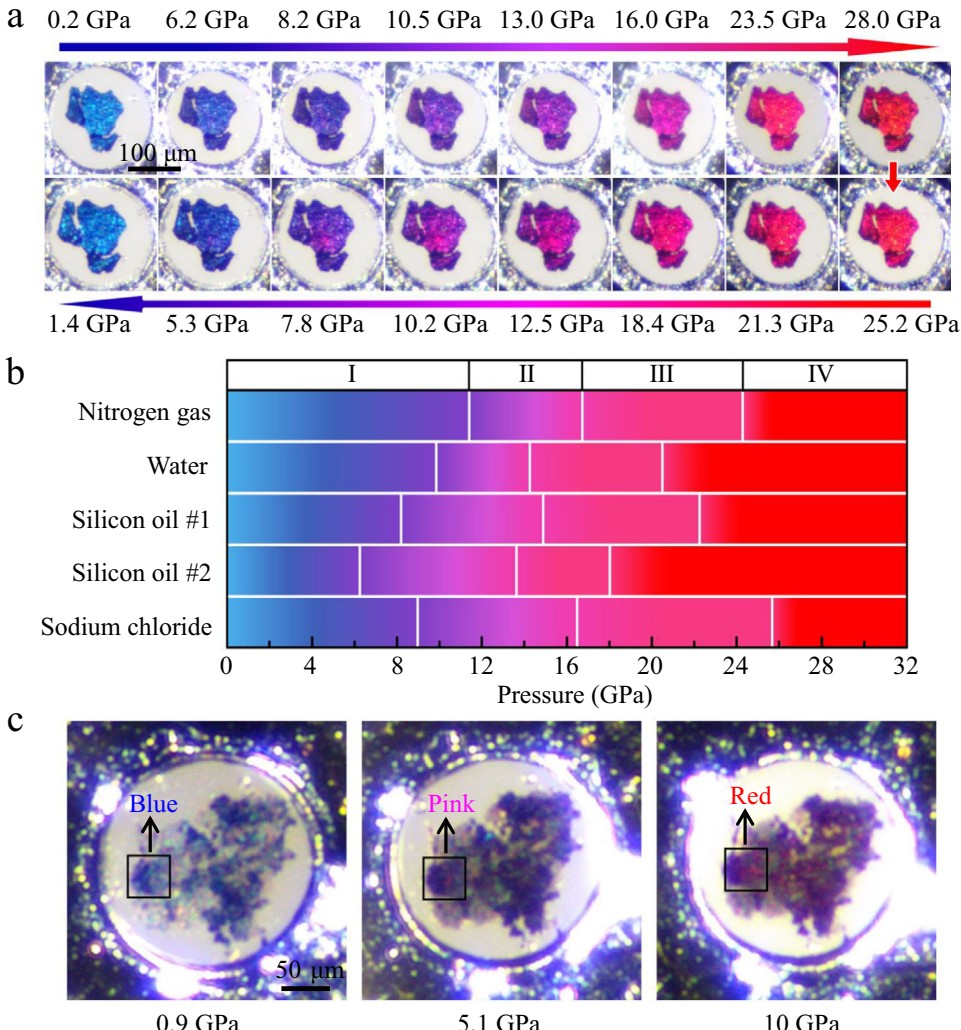

**Fig. 2 | Evolution of color changes of nitrogen doped lutetium hydrides with varying pressures. a** Optical images of pressure-induced color changes of the produced LuH$_{2\pm x}$N$_y$ samples during compression (upper images) and decompression (down images) processes in a DAC chamber with silicon oil as pressure medium. **b** Phase diagram of phase I (blue), phase II (purple), phase III (pink) and phase IV (red) of the LuH$_{2\pm x}$N$_y$ samples in different pressure media. Two runs were carried out in silicon oil medium with Re-gasket thickness of 32 μm and 44 μm, respectively. The sample chamber with thick gasket underwent expansion, creating shear stresses, which helped reduce the critical pressures of color changes in the LuH$_{2\pm x}$N$_y$ sample (Supplementary Fig. 10). **c** Optical images of the samples showcasing reduced critical pressures of the LuH$_{2\pm x}$N$_y$ samples mixed with nanodiamonds by internal stress in the NaCl pressure medium.

pink phase are consistent with the appearance of the claimed superconductivity[1].

These results offer strong evidence that the pressure conditions have major effects on the chemical interaction between nitrogen and the structural frame of lutetium hydrides in the pink phase region. Our further theoretical calculations provide evidence that the incorporation, distribution of N/NH can significantly influence the electron transfer between the existing vacancies and Lu-H framework (see Supplementary Figs. 16–19). This may be a key reason for the gradual color evolution from its original blue to the ultimate red under compression.

**Electrical transport measurements of the LuH$_{2\pm x}$N$_y$ samples**
We performed electrical transport measurements in the four phases over the temperature range of 1.8–300 K. Figure 4a show that LuH$_{2\pm x}$N$_y$ samples exhibit typical metallic behaviors at ambient pressure. We further measured the temperature dependent resistance under pressure in five runs. Figure 4b shows pressure evolution of resistance at room temperature. In sample-I, the resistance in run 1

decreases with rising pressure up to 8.3 GPa and drops by an order of magnitude between 8.3 and 13.3 GPa, then stays nearly constant to the highest measured pressure of 29.9 GPa. In run 2, under rising pressure, the resistance first increases until pressure reaches 3.7 GPa, then gradually decreases up to pressure of 30.9 GPa (see Supplementary Fig. 20). Such diverse behaviors of resistance under pressure likely stem from the variation of the nitrogen contents taken from different parts of the original synthesized sample. In sample-II, the resistance gradually decreases up to pressure of 38.2 GPa (run 3–5), similar to that observed in run 2 of sample-I. In Fig. 4c, d, we present the resistance data during the cooling-down measurement, and the results show no sign of superconductivity in all four phases of LuH$_{2\pm x}$N$_y$ samples (blue to red) from 300 K to 1.8 K. Our further experiments also preclude the possibility of the reported superconductivity[1] from the secondary phase of LuN$_{1-\delta}$H$_\varepsilon$ in ref. 1. up to 31.29 GPa (see Supplementary Fig. 21).

The measured resistance curve should exhibit the same trend of variation with changing temperature during cooling-down or warming-up process with same cooling/warming rate if no

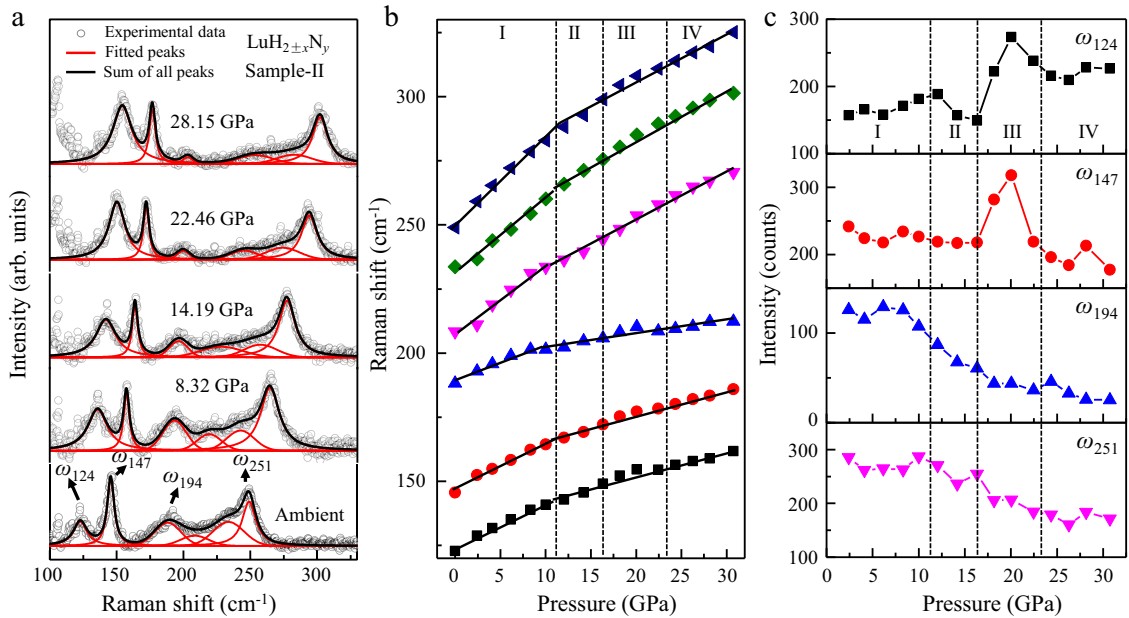

**Fig. 3 | In-situ Raman spectrum of nitrogen doped lutetium hydrides at changing pressures. a** Typical pressure-dependent Raman spectra of the produced LuH$_{2\pm x}$N$_y$ (sample-II) with a 633 nm laser excitation. The spectral curves (cycles) are deconvoluted by Gaussian fitting (solid lines). **b** Pressure-dependent shift of the fitted Raman peaks. **c** Pressure-dependent intensity of the Raman peaks of 124 cm$^{-1}$, 147 cm$^{-1}$, 194 cm$^{-1}$, and 251 cm$^{-1}$.

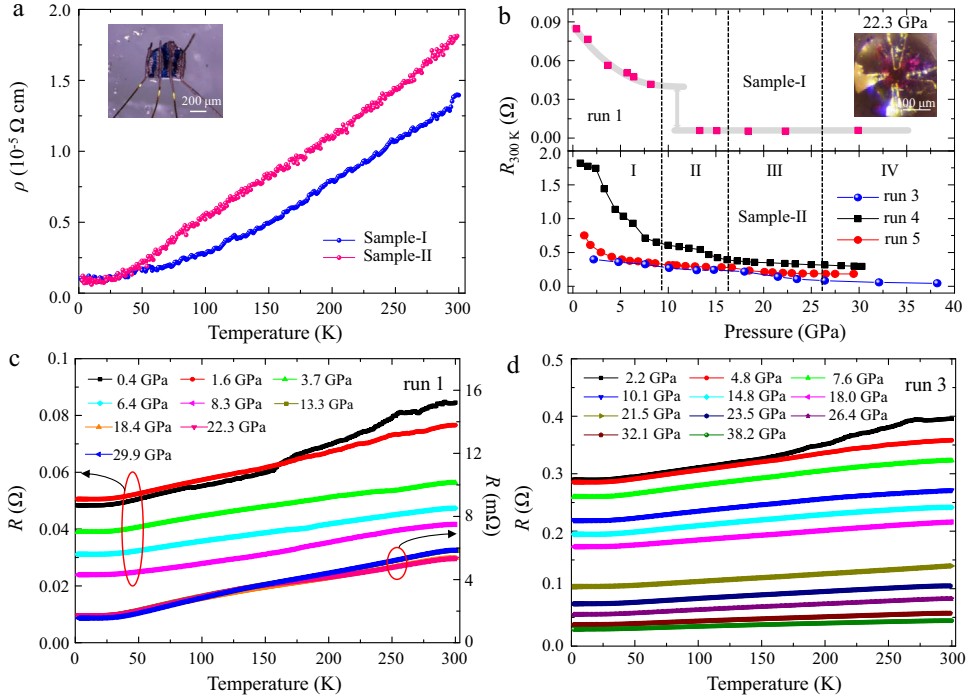

**Fig. 4 | Temperature dependent electrical resistance of the LuH$_{2\pm x}$N$_y$ samples at changing pressures. a** Temperature dependence of resistivity at ambient pressure. Top inset shows the optical micrograph of LuH$_{2\pm x}$N$_y$ sample with gold electrodes attached by silver paste. **b** Room temperature resistance of sample-I (top) and sample-II (bottom) at different pressures. The shading line is a guide to the eyes. **c**, **d** Evolution of the resistance with pressure at temperature ranging from 300 K to 1.8 K during cooling down process of sample-I (run 1) and sample-II (run 3), respectively.

temperature-driven phase transition occurs. Generally, data collected during warming-up are used for analysis because of the more homogeneous thermal equilibrium than in the cooling-down measurements. Surprisingly, we observed contrasting behaviors in the measured $R$(T) curves of LuH$_{2\pm x}$N$_y$ samples during the cooling-down and warming-up electrical measurements (Supplementary Figs. 22-24), and the contrast

is especially pronounced in the pink phase (III) that was claimed to host near-ambient superconductivity[1].

It is interesting to note in Fig. 5a that, in the region of purple phase II (13.3 GPa) and pink phase III (22.3 GPa), a plateau in the resistance curve for sample-I develops and grows with rising pressure starting at about 260 K during the warming-up

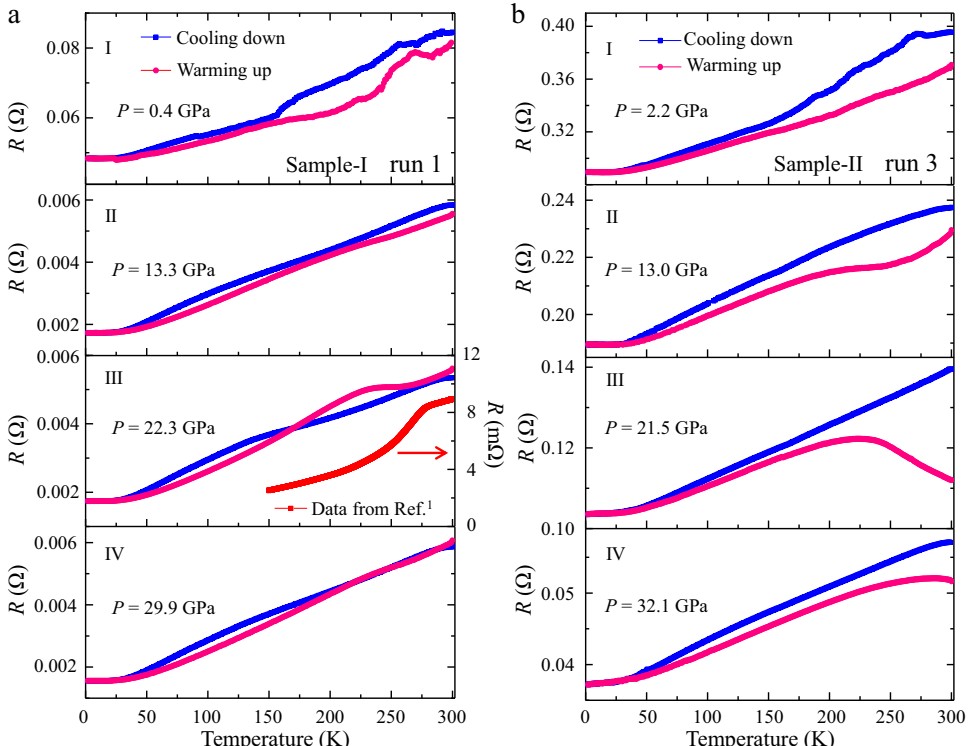

**Fig. 5 | Contrasting $R(T)$ curves obtained in cooling-down and warming-up measurements. a** Temperature dependence of resistance of sample-I (run 1) for phase I at 0.4 GPa, phase II at 13.3 GPa, phase III at 22.3 GPa and phase IV at 29.9 GPa. **b** Temperature dependence of resistance of sample-II (run 3) for phase I at 2.2 GPa, phase II at 13.0 GPa, phase III at 21.5 GPa and phase IV at 32.1 GPa. The raw resistance data on the pink phase (taken at lower pressure) from the recent work reporting on ambient superconductivity[1] are included (panel III) to show a similar plateau observed in our warming-up curve, despite a shift in temperature where the turning point of the curve is located.

measurements, followed by a relatively sharp drop with reducing temperature; however, no zero-resistance state was observed down to 1.8 K. Such a feature significantly weakens when the sample turns to red color (phase IV). Such behaviors are possibly caused by pressure- or/and thermally driven influence on the electron transfer in $LuH_{2\pm x}N_y$ due to the incorporation of N/NH and hydrogen vacancies[48,49], as indicated by the Raman spectra shown in Fig. 3c. The onset temperature of the resistance anomaly in $LuH_{2\pm x}N_y$ is sensitive to pressures and rises in the pink phase with increasing pressure (see Supplementary Fig. 25).

Similar resistance anomalies are also observed in sample-II (Fig. 5b) with the same N level as that in the claimed near-ambient superconductor[1]. The plateau in the resistance curve is observed even in the purple phase (II) at 13.0 GPa, and the hump structure becomes more pronounced (21.5 GPa) through the pink phase III and persists into the red phase IV above 32.1 GPa. We also measured $R(T)$ with an applied magnetic field of 3 T at 26.1 GPa in run 2, the results show no shift in the critical pressure marking the turning point in the resistance curve, and this lack of response of resistance to applied magnetic field excludes the possibility of a superconducting transition for this abnormal behavior (Supplementary Fig. 23 j).

In conclusion, we have synthesized high-purity nitrogen doped lutetium hydrides ($LuH_{2\pm x}N_y$) using HPHT method, and the obtained samples exhibit the same crystal structure, composition and overall color changing trends as those of the recently reported near-ambient superconductor[1]. Our experimental and theoretical results provide convincing evidence of nitrogen incorporation in the $LuH_{2\pm x}N_y$ samples. Our extensive and systematic resistance measurements indicate that there is no superconducting transition in all the blue, purple, pink and red phases of $LuH_{2\pm x}N_y$ samples at

temperatures of 1.8-300 K and pressures from 0.4 GPa to 38 GPa. These results demonstrate that the pressure/temperature-driven electron redistribution by incorporation of nitrogen/vacancies and its interaction with the $LuH_2$ framework plays a crucial role in the remarkable visual color changes and anomalous electrical resistance behaviors seen in the experiments. The present work has shown unambiguous evidence that there is no correlation between the pink phase and near-ambient superconductivity in nitrogen doped lutetium hydrides, and further efforts should focus on elucidating the origin of the intriguing emergent N related bonding changes that may drive the sample color changes and the associated resistance anomalies.

## Methods

### Starting materials and HPHT synthesis

The polycrystalline samples of $LuH_{2\pm x}N_y$ were synthesized using a China-type cubic-type high pressure apparatus. The high-pressure cell used in this study has two layers (layer 1 and layer 2) that were separated by a BN thin plate, as depicted by a schematic diagram in Supplementary Fig. 1. In layer 1, the Lu pieces (99.9 *wt*.% purity) with silver color were placed as the precursors. Layer 2 was filled with the mixture of $NH_4Cl$ (Aladdin 99.99 *wt*.% purity) and $CaH_2$ (Aladdin 98.5 *wt*.% purity) in a molar ratio of 2:8 that was used as the source of nitrogen and hydrogen[15]. Lu pieces were commercially purchased from Hebei Rechen New Material Technology Co., Ltd. The cell assembly was then heated at 773 K for 5 ~ 7 hours under pressures of 3 GPa, followed by rapid cooling to room temperature. Finally, $LuH_{2\pm x}N_y$ samples were obtained after pressure release. After the HPHT experiments, we carefully removed the surrounding *h*-BN materials and remaining lutetium on the surfaces (Supplementary Fig. 1b), and then retrieved

the final products for characterization and further high-pressure experiments. The obtained HPHT products have a dominated shining blue phase and a very tiny amount of purple phase on the surface. We focus our study on the most interesting blue phase (Supplementary Fig. 1c). The $LuN_{1-\delta}H_\varepsilon$ samples were synthesized with the same conditions as $LuH_{2\pm x}N_y$. The starting materials are LuN powders (99.999 $wt.\%$ purity) that are commercially available from Shanghai Yien Chemical Reagent Co., Ltd.

## Sample characterization
The crystal structure of produced samples was examined using an X-ray diffractometer (XRD) with Cu-Kα radiation ($\lambda = 1.5406$ Å, PANalytical X'pert3, Holland). High resolution transmission electron microscopy (HRTEM) images were obtained on a JEM2100 Plus transmission electron microscope at an acceleration voltage of 200 kV. Elemental analysis was made by a scanning electron microscope (SEM, Zeiss Sigma 500) equipped with energy dispersive x-ray (EDX) spectroscopy probe using an accelerating voltage of 5 kV. X-ray photoelectron spectroscopy (XPS) spectrums were taken by a Thermo Scientific (ESCALAB, 250Xi). During the experiments, the original surface of the studied samples was removed by etching techniques in order to avoiding the contamination of water and oxygen in air. Raman experiments were carried out on a high-resolution Raman spectrometer (Horiba, LabRAM HR revolution) with the excitation wavelength of 532 nm and 633 nm (grating: 1800 g/mm). The electrical transport measurement was carried out on a Quantum Design Physical Property Measurement System (PPMS).

## In-situ high pressure measurements
High pressure resistivity measurements were conducted in a screw-pressure-type diamond-anvil-cell (DAC) made of non-magnetic Be-Cu alloy. A mixture of $c$-BN powder and epoxy was used as the insulating coating for the rhenium gaskets, which was pre-indented to 30 μm in thickness. Several pieces of $LuH_{2\pm x}N_y$ grains with shinning blue color were selected under a microscope and loaded into the gasket hole. The Pt electrodes were attached to the sample with a four-probe van der Pauw method. NaCl was used as the pressure medium. Also, water, silicone oil, and nitrogen gas were used in the experiments of pressure induced color change. High-quality ruby balls ~10 μm in size were used for pressure calibration. The Raman peak position is taken from the center of the full width at half-maximum (FWHM), while the peak intensity is calculated after subtracting the background by considering the symmetry and FWHM of the peak.

## Theoretical calculations
We used the $Fm\bar{3}m$ $LuH_2$ as the prototype, and constructed $LuH_{1.875}$ with H vacancies in a $2 \times 2 \times 2$ supercell containing 96 atoms. Then one N atom and one NH cluster are inserted into interstitial sites to simulate N doped $LuH_{1.875}$ systems, denoted as $Lu_{32}H_{60}$-N and $Lu_{32}H_{60}$-NH, respectively. The precise geometry relaxations and electronic structure calculations were carried out within the framework of density functional theory (DFT) as implemented in the Vienna ab initio simulation package (VASP)[50]. The electron–ion interactions are described by the generalized gradient approximation of Perdew-Burke-Ernzerhof (PBE)[51] functional. The projector-augmented wave (PAW)[52] pseudopotentials with the valence electrons $5p^65d^146s^2$ for Lu, $1s^1$ for H, and $2s^22p^3$ for N were used. The energy cutoff of 550 eV and a Monkhorst-Pack scheme[53] with a dense k-point spacing of $2\pi \times 0.03$ Å$^{-1}$ were chosen in all the calculations. The electron localization function (ELF)[54] was calculated to illustrate the electron distribution and chemical bonding. Phonon and Raman spectra were calculated within the PHONOPY code[55].

## Data availability
All data generated or analyzed in this study are included in this article and its Supplementary Information file. Additional data are available from the corresponding author upon request.

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

## Acknowledgements

X.L. would like to thank the whole HPPMS team for the hard work. The authors would like to thank Xiaofeng Xu for fruitful discussion. This work was partly supported by the National Natural Science Foundation of China (Grant Nos. 12204265 X.X., 11974208 X.L., 52172212 X.C., and 52102335 C.W.), the Natural Science Foundation of Shandong Province (Grant Nos. ZR2023JQ001 X.L., ZR2020YQ05 X.L., ZR2022QA040 X.X., and ZR2021YQ03 X.C.), the Higher Educational Youth Innovation Science and Technology Program of Shandong Province (2022KJ183 Y.L.), the Young Scientists of Taishan Scholarship (tsqn202211128 X.L. and tsqn202306184 X.C.), and the Strategic Priority Research Program (B) of the Chinese Academy of Sciences (Grant No. XDB25000000 Z.S.).

## Author contributions

X.L. and X.X. designed the experiments and gathered data. X.L., X.X., C.C., and Y.M. analyzed the data, organized the results and wrote the manuscript. X.X., L.Y., S.H., and X.L. synthesized the samples. X.X, C.W., J.X., M.Z., C.Z., and X.Z. performed sample characterization and high-pressure measurements. Y.L., X.C. and Y.Z. performed the theoretical calculations and analysis. B.Y., J.G., and Z.S. provided discussions. All authors contributed to editing and improving the manuscript.

## Competing interests

The authors declare no competing interests.
