## [Peer Review File · Nature Communications]

Observation of non-superconducting phase changes in nitrogen doped lutetium hydridesEditorial Note: This manuscript has been previously reviewed at another journal that is not operating a transparent peer review scheme. This document only contains reviewer comments and rebuttal letters for versions considered at Nature Communications.

REVIEWER COMMENTS

Reviewer #2 (Remarks to the Author):

This manuscript reports the synthesis of high-purity nitrogen-doped lutetium hydride $\text{LuH}_{2\pm x}\text{N}_y$ with the same structure and composition as in the claimed near-ambient superconductor by Ref. 1. By the electrical resistance measurements, the authors find no evidence of the superconductivity. Instead, they observed a hump appears in the Resistance vs. Temperature curves near 200 K during the warming-up measurements. The authors point out that this hump may be misinterpreted as the indication of superconducting transition in Ref. 1.

This manuscript also reports the detailed investigation of the pressure-induced color change, hence the phase transitions, concerning crystal structure and electron redistribution by incorporating nitrogen/vacancies and its interaction with the LuH_2 framework.

Regarding the 3rd manuscript submitted to Nature, two reviewers raised concerns regarding the phases studied by the present experimental study. The first one is that the studied phase may differ from that studied in Ref. 1. Others were the LuN and $\text{LuN}_{1-\delta}\text{H}_\epsilon$ as impurities and potential nitrogen sources under compression.

The authors addressed the first concern by analyzing the Raman scattering data and investigating the internal stress effect. Their analysis reveals that the phase studied is the same as Ref. 1.

The authors also addressed the second concern by investigating the high-pressure responses of LuN and $\text{LuN}_{1-\delta}\text{H}_\epsilon$ by Raman scattering and XRD measurements.

The current manuscript provides significant insight into what happens in this topical material $\text{LuH}_{2\pm x}\text{N}_y$, not only at high pressures but also at varying temperatures.

This study's experimental results and conclusions are significant and valuable to readers interested in superconductivity and superconductor synthesis. This reviewer can recommend it for publication.

Reviewer #3 (Remarks to the Author):

The revised manuscript by Xing et al., submitted to NatComm, has addressed several important issues raised by reviewers. However, two of these issues have not been satisfactorily explained.

1. XPS spectra. I do not see what is wrong in my interpretation of your spectra. For LuN phase you see two peaks, with areas 1:2. You claim: "...indicating that the two chemical bonds of nitrogen atoms are binding with Lu and H atoms in a ratio of 1:2, rather than the ratio of Lu(N) and NH or NH_2 phases are 1:2 in the studied samples". First, this sentence is totally unclear. What are the "two

chemical bonds of nitrogen"? Secondly, you confirmed my surmise that "similar XPS spectra .. were observed for Li_2NH and LiNH_2 " (JAACmpds, 2022). This means that I was right about "protic hydrogen" in LuN samples. Moreover, since there is A LOT of N in LuN sample, and the peak of N bound to H is substantial, there MUST be a lot of protic H in these samples, as well. Where is the misunderstanding? This issue is very important since you also report XPS spectra for N-poor LuH₂ phases, and you try to interpret these spectra. Most weirdly, S/N ratio for N1s peaks is better for N-poor than N-rich samples! One would expect tiny if observable N1s signals from N-poor LuH₂. Everything remains totally unclear.

2. Raman spectra

You claim you have now measured Raman spectra in the 2000-3600 cm^{-1} region, since I suggested to monitor the NH stretching region. However, you show in your response and also in Supplement the N-N stretch (very narrow) region only. First, your discussion of a mere 2 cm^{-1} shift with respect to gaseous N₂ is purely speculative; such minor shift may come from zillion different effects. Secondly, the fact that you observe N-N stretch means that there is some triply bonded N₂ in LuN samples??? This does not make any sense, s LuN is an ionic compound. Maybe you measure some N₂ chemisorbed on its surface. Third, you totally avoid showing the NH stretching region, so you do not permit the reader to understand BOTH XPS and Raman spectra for NH species present in the samples.

If these two key issues - which may clarify the chemical state of N in your samples - are not resolved, I would NOT be in favour of publishing this work. Interpretations are too messy.

Authors' replies to the Reviewer comments

REVIEWER REPORTS

Reviewer comments in italics

Page numbers refer to the "track-changes on" version of the manuscript

Reply to the Report of Reviewer #2

Comments from Reviewer #2:

This manuscript reports the synthesis of high-purity nitrogen-doped lutetium hydride $\text{LuH}_{2\pm x}\text{N}_y$ with the same structure and composition as in the claimed near-ambient superconductor by Ref. 1. By the electrical resistance measurements, the authors find no evidence of the superconductivity. Instead, they observed a hump appears in the Resistance vs. Temperature curves near 200 K during the warming-up measurements. The authors point out that this hump may be misinterpreted as the indication of superconducting transition in Ref. 1.

This manuscript also reports the detailed investigation of the pressure-induced color change, hence the phase transitions, concerning crystal structure and electron redistribution by incorporating nitrogen/vacancies and its interaction with the LuH_2 framework.

Regarding the 3rd manuscript submitted to Nature, two reviewers raised concerns regarding the phases studied by the present experimental study. The first one is that the studied phase may differ from that studied in Ref. 1. Others were the LuN and $\text{LuN}_{1-\delta}\text{H}_\epsilon$ as impurities and potential nitrogen sources under compression.

The authors addressed the first concern by analyzing the Raman scattering data and investigating the internal stress effect. Their analysis reveals that the phase studied is the same as Ref. 1. The authors also addressed the second concern by investigating the high-pressure responses of LuN and $\text{LuN}_{1-\delta}\text{H}_\epsilon$ by Raman scattering and XRD measurements.

The current manuscript provides significant insight into what happens in this topical material $\text{LuH}_{2\pm x}\text{N}_y$, not only at high pressures but also at varying temperatures.

This study's experimental results and conclusions are significant and valuable to readers interested in superconductivity and superconductor synthesis. This reviewer can recommend it for publication.

Reply: We thank the reviewer for a careful assessment of the work reported in our manuscript and the recommendation for its publication in *Nature Communications*.

Reply to the Report of Reviewer #3

Comments from Reviewer #3:

The revised manuscript by Xing et al., submitted to NatComm, has addressed several important issues raised by reviewers. However, two of these issues have not been satisfactorily explained.

1. XPS spectra. I do not see what is wrong in my interpretation of your spectra. For LuN phase you see two peaks, with areas 1:2. You claim: "...indicating that the two chemical bonds of nitrogen atoms are binding with Lu and H atoms in a ratio of 1:2, rather than the ratio of Lu(N) and NH or NH₂ phases are 1:2 in the studied samples". First, this sentence is totally unclear. What are the "two chemical bonds of nitrogen"? Secondly, you confirmed my surmise that "similar XPS spectra .. were observed for Li₂NH and LiNH₂" (JAACmpds, 2022). This means that I was right about "protic hydrogen" in LuN samples. Moreover, since there is A LOT of N in LuN sample, and the peak of N bound to H is substantial, there MUST be a lot of protic H in these samples, as well. Where is the misunderstanding? This issue is very important since you also report XPS spectra for N-poor LuH₂ phases, and you try to interpret these spectra. Most weirdly, S/N ratio for N1s peaks is better for N-poor than N-rich samples! One would expect tiny if observable N1s signals from N-poor LuH₂. Everything remains totally unclear.

Reply: We appreciate the reviewer's constructive comments. Our reply is given below.

First, "two chemical bonds of nitrogen" refers to the two kinds of chemical bonds formed by the incorporated nitrogen atoms in the LuH₂ framework as Lu-N bonds and Lu-H bonds. These two kinds of chemical bonds involving nitrogen could not be directly reflected in the XRD and Raman spectra in the new phases as Lu(N)_{2/3}(NH or NH₂)_{1/3}.

Second, we agree with the reviewer that the "protic hydrogen" as NH or NH₂ species play important roles in the phase transition of LuH₂. We have performed additional calculations to analyze the chemical bonding of nitrogen for further understanding of the effect of N-H incorporation in LuH₂. The following changes have been made in revised manuscript.

Lines 139-141: "*while the peaks at the lower wavenumbers of 124 cm⁻¹ and 147 cm⁻¹ are only observed in nitrogen doped lutetium hydrides that can be assigned to Lu-NH and Lu-N related vibrational modes (Supplementary Figs. 2 and 3)*".

Lines 147-149: "*This suggests that the dominant bonding configuration for nitrogen atoms and NH species is N-H bonding in the produced LuH_{2±x}N_y samples⁴¹⁻⁴³*".

Lines 226-228: "*It is interesting to note that the pressure driven changes of the intensity for Lu-N and Lu-N-H related peaks in the pink phase are consistent with the appearance of the claimed superconductivity¹*".

Third, the S/N ratio for N 1s peaks is not only dependent on the nitrogen content but also the crystalline quality of the samples, as well as signal accumulation during XPS measurements. We performed hundreds of tests to reduce the signals from N-poor LuH₂ samples. The accumulation in the obtained XPS spectra reflect averaged data, which can provide reliable results to analyze chemical bonding environments in the N-rich LuN samples.

2. Raman spectra. You claim you have now measured Raman spectra in the 2000-3600 cm⁻¹ region, since I suggested to monitor the NH stretching region. However, you show in your response and also in Supplement the N-N stretch (very narrow) region only. First, your discussion of a mere 2 cm⁻¹ shift with respect to gaseous N₂ is purely speculative; such minor shift may come from zillion different effects. Secondly, the fact that you observe N-N stretch means that there is some triply bonded N₂ in LuN samples??? This does not make any sense, as LuN is an ionic compound. Maybe you measure some N₂ chemisorbed on its surface. Third, you totally avoid showing the NH stretching region, so you do not permit the reader to understand BOTH XPS and Raman spectra for NH species present in the samples.

Reply: Considering the NH stretching region is usually shown in the high wavenumber region, we measured the Raman spectra in the 2000-3600 cm⁻¹ region; but unfortunately, no NH stretching modes can be clearly detected throughout the whole measured region. Only a sharp peak for the possible N-N stretch mode at 2328 and 2324.5 cm⁻¹ can be observed in the N-rich and N poor samples. To compare the difference of the N-related peaks, we showed the region of 2250-2400 cm⁻¹. Please see the attached data for the whole spectra.

Attached figure. The Raman spectra of LuH_{2+δ}N_γ and LuN_{1-δ}H_ε samples at higher wavenumber regime. The marked jump in the red cycle is derived from the grating change during the Raman measurements.

We also agree with the reviewer that the possibility of “ N_2 chemisorbed on its surface” cannot be excluded during the analysis of Raman measurements in air. Accordingly, we have deleted this data in our revised Supplementary Information.

To investigate the effect of N-H species in these stretching modes, we performed further calculations to compare computed and measured Raman spectra in our revised manuscript. We found the NH species are stable in the LuH_2 frameworks, while the NH_2 species tend to induce structural instability. Our theoretical results demonstrate that the “protic hydrogen” bonded with nitrogen as NH species has significant influence on the Raman stretching modes in the lower wavenumber region, especially below 150 cm^{-1} . Please see the attached new supplementary Fig. 3 in our revised Supplementary Information.

Supplementary Fig. 3 | Theoretical simulation of crystal structure and Raman spectra. a, Structures of the pristine LuH_2 compound, with H vacancies ($LuH_{1.875}$), and plus one N interstitial ($Lu_{32}H_{60-N}$) and one NH cluster ($Lu_{32}H_{60-NH}$) nearby vacancies (from top to bottom). **b** and **c**. Comparison of experimental and calculated Raman spectra. Since the cubic structure with $Fm\bar{3}m$ space group has high symmetry, low energy

Raman modes arising from the reduced symmetry of the LuH₂ framework are induced by the H vacancies in the lattice. The Raman modes become rich and strong when N atoms are incorporated in Lu₃₂H₆₀-N, resulting in further reduction of the symmetry of the structure, consistent with the experimental results. Interestingly, a Lu-N related vibration at 154 cm⁻¹ (marked as red line) in Lu₃₂H₆₀-N contributes to the presence of the modes at 147 cm⁻¹ in the measured LuH_{2+x}N_y samples¹. The incorporated NH species in Lu₃₂H₆₀-NH play significant roles in the lower wavenumber region, inducing the presence of three Raman vibrations as 104.5, 112.3 and 120.6 cm⁻¹ (marked as blue lines), which is consistent with the observation of a broad band at 124 cm⁻¹ in the measured Raman spectra of LuH_{2+x}N_y samples¹. While the origin of the two peaks at 194 cm⁻¹ and 251 cm⁻¹ in the measured LuH_{2+x}N_y samples (both shown in recently reported LuH₂ samples²) remains uncertain, it is reasonable to attribute them to other factors inducing the reduction of structural symmetry by vacancies related defects or interstitial hydrogen atoms.

On the basis of our calculated and measured Raman spectra, the two characteristic peaks at 124 cm⁻¹ and 147 cm⁻¹ of LuH_{2+x}N_y (Fig. 1f and Fig.3) can be assigned to the stretching modes of Lu-NH and Lu-N bonds, respectively. We have revised the related discussion of our revised manuscript.

If these two key issues - which may clarify the chemical state of N in your samples - are not resolved, I would NOT be in favour of publishing this work. Interpretations are too messy.

Reply: We thank the reviewer for making the constructive comments, which have helped us to improve our analysis of the measured XPS data for better understanding the chemical state of N in our produced samples. We also responded to the reviewer's comments and analyzed the effect of the "protic hydrogen" as NH species in the Raman spectra and electron transfer of the LuH₂ under compression. New data have been added in supplementary Fig. 3c, Fig. 15d and Fig. 18b. Related discussions also were amended in the revised manuscript.

We feel that we have addressed the concerns raised by the reviewer and hope that our latest revised manuscript is now suitable for publication in *Nature Communications*.

REVIEWERS' COMMENTS

Reviewer #3 (Remarks to the Author):

The authors have finally taken seriously all remarks from this reviewer. They performed additional experiments and calculations, and gave satisfactory responses. While not everything is still clear, I believe the manuscript may be accepted for publication and all remaining question marks may be left for future research. I suggest, however, that the already measured Raman spectra in the NH stretching region are added to the supplement to testify that the amount of NH(2-) species is small.

Authors' replies to the Reviewer comments

REVIEWER REPORTS

Reviewer comments in italics

Page numbers refer to the "track-changes on" version of the manuscript

Reply to the Report of Reviewer #3

Comments from Reviewer #3:

The authors have finally taken seriously all remarks from this reviewer. They performed additional experiments and calculations, and gave satisfactory responses. While not everything is still clear, I believe the manuscript may be accepted for publication and all remaining question marks may be left for future research. I suggest, however, that the already measured Raman spectra in the NH stretching region are added to the supplement to testify that the amount of NH(2-) species is small.

Reply: We thank the Reviewer for his/her recommendation for the publication of our manuscript in *Nature Communications*. According to the Reviewer's suggestion, we have added the Raman spectra measured at high wavenumber region as new Supplementary Fig. 3 in the revised Supplementary Information (Page 4).